# Systematic Review of Intervention Studies Aiming at Reducing Inequality in Dental Caries among Children

**DOI:** 10.3390/ijerph18031300

**Published:** 2021-02-01

**Authors:** Anqi Shen, Eduardo Bernabé, Wael Sabbah

**Affiliations:** 1Department of Preventive Dentistry, Beijing Stomatology Hospital, Capital Medical University, 4th Tiantanxili, Dongcheng District, Beijing 100050, China; 2Dental Public Health Group, Faculty of Dentistry, Oral & Craniofacial Sciences, King’s College London, London SE5 9RS, UK; Eduardo.bernabe@kcl.ac.uk (E.B.); wael.sabbah@kcl.ac.uk (W.S.)

**Keywords:** child, dental caries, inequalities, socioeconomic factors, systematic review

## Abstract

(1) Background: The objective is to systematically review the evidence on intervention programs aiming at reducing inequality in dental caries among children. (2) Methods: Two independent investigators searched MEDLINE, Cochrane library, and Ovid up to December 2020 to identify intervention studies assessing the impact on socioeconomic inequalities in dental caries among children. The interventions included any health promotion/preventive intervention aiming at reducing caries among children across different socioeconomic groups. Comparison groups included children with alternative or no intervention. Cochrane criteria were used to assess interventional studies for risk of bias. (3) Results: After removal of duplicate studies, 1235 articles were retained. Out of 43 relevant papers, 13 articles were identified and used in qualitative synthesis, and reported quantifiable outcomes. The included studies varied in measurements of interventions, sample size, age groups, and follow-up time. Five studies assessed oral health promotion or health-education, four assessed topical fluorides, and four assessed water fluoridation. Interventions targeting the whole population showed a consistent reduction of socioeconomic inequalities in dental caries among children. (4) Conclusion: The quality of included papers was moderate. High heterogeneity did not allow aggregation of the findings. The overall findings suggest that whole population interventions such as water fluoridation are more likely to reduce inequalities in children’s caries than target population and individual interventions.

## 1. Introduction

Dental caries is considered one of the most prevalent chronic diseases worldwide [1]. The percentage of children among disadvantaged families who suffered from early childhood caries was reported as 70% [2]. Despite the overall decline in dental caries over the past few decades [3], inequalities in oral health and particularly in dental caries continue to exist and present a challenge to health promotion interventions [4]. Aside from the known proximal risk factors for dental caries [5], socioeconomic position (commonly measured by family income, occupation, parental education, and area deprivation), defined by social processes, materialistic conditions, and life changes over time, subsequently lead to oral health inequalities [6]. Low socioeconomic position, low family income, and low maternal educational levels are also related to proximal risk factors for dental caries such as poor access to dental services, inadequate use of oral hygiene products, poorer oral health knowledge, and oral hygiene [5], which result in high prevalence and severity of dental caries [7].

Among different preventive interventions, fluoride is found to be an effective and non-invasive agent for the management of dental caries, particularly water fluoridation and fluoride toothpaste, gel, and mouthwash, which are commonly incorporated into public health interventions to reduce the prevalence of dental caries [8]. On the other hand, oral health education as a health promotion intervention aims at changing behavior and subsequently reducing the burden of caries. However, several studies argued that, while oral health education could be effective in improving knowledge, it is not effective in producing sustainable behavior change or preventing dental caries [9]. However, health education appears to increase inequalities in dental caries as it has a better impact on children in higher socioeconomic areas [10,11].

There are few systematic reviews or reviews that assessed the effectiveness of fluoride vanish or oral health promotion interventions in preventing dental caries [12,13,14]. However, there is no previous systematic review of intervention studies aiming at reducing inequality in caries among children. The current review aims at filling this gap in the literature.

While a few studies found intervention programs that reduced inequality in dental caries among children [12,15,16,17], others have argued that intervention programs increased or had no effect on inequality [18,19,20]. Given the inconclusive views on the effect different interventions on inequality in dental caries, we conducted a systematic review of the available intervention studies that explored the effect on reducing inequality in dental caries. The aim of this study was to systematically review the evidence on intervention programs aiming at reducing inequality in dental caries among children.

## 2. Materials and Methods

This systematic review followed the Cochrane Handbook guidelines for systematic reviews of intervention studies [21]. This review was registered in PROSPERO (registration number: CRD42020179342).

### 2.1. Eligibility Criteria

PICO’s format was used as criteria for eligibility; that is, ‘P’, participants were children or adolescents aged up to 18 years; ‘I’, intervention groups included children exposed to any form of clinical intervention or oral health education/oral health promotion activity; ‘C’, comparison groups included children with no intervention or with an alternative intervention; and ‘O’, studies that assessed or reported reduced inequalities in dental caries among children.

Inclusion criteria:

Intervention studies: (a) randomized controlled trails (RCTs), (b) controlled trails without randomization, and (c) interventions with before and after comparison.

Exclusion criteria:

(1) Observational studies (cross-sectional and longitudinal studies) and reviews were excluded. (2) Studies on adults. (3) Studies that did not assess dental caries as an outcome. (4) Studies that did not assess socioeconomic inequality in dental caries.

### 2.2. Study Selection

Two independent reviewers (A.S. and W.S.) conducted the literature search using three databases (Pubmed, Cochrane library, and Ovid) up to December 2020. Search terms are shown in Appendix A. Published and accessible papers were considered in the literature review. The authors also contacted and asked for grey literature. Papers were filtered by their title and abstracts for relevance, respectively. Finally, papers were included by reading the entire articles (shown in Figure 1). All references were obtained in software Endnote X9 Endnote (Endnote Software: Release X9. Philadelphia, PA, USA).

### 2.3. Data Extraction

Following agreement between the two reviewers on studies that met inclusion criteria, data pertaining to study design, including authors, year of publication, country, and participants’ characteristics (sample size, age, follow-up duration, intervention test and control group, outcomes, results, and conclusions) were extracted from the included papers. The studies’ characteristics are summarized in Appendix A.

### 2.4. Risk of Bias Assessment

Two investigators independently assessed the included articles. These studies were evaluated to assess the quality of methodology. The included intervention studies were assessed for risk of bias following the Cochrane criteria of systematic review for intervention studies [21]. These included random sequence generation, allocation concealment, blinding of participants and personnel, blinding of outcome assessment, incomplete outcome data, selective reporting, and other bias. The quality of included studies is shown in Figure 2.

### 2.5. Data Synthesis

Different interventions were presented in the included studies, such as oral health promotion, oral health education, motivational interviews, topical fluorides, dental sealants, and water fluoridation. Owing to the high level of heterogeneity among the included studies, meta-analysis was not conducted. Instead, a qualitative synthesis of the studies was conducted. The included studies’ characteristics are summarized in Appendix A, and quality assessment of the included studies is shown in Figure 2.

## 3. Results

### 3.1. Studies’ Selection

Overall, papers were identified through three databases (841 from PubMed, 544 from Cochrane library, and 299 from Ovid). After removal of duplicate studies, 1235 articles were retained. Two investigators independently screened the titles of the retained articles. After filtering titles, an additional 66 papers were deemed irrelevant and excluded after reviewing the abstract. The final decision about whether a study met the inclusion criteria was made based on the full-text and after discussion between investigators. Out of 43 papers included at the full text stage, an additional 30 studies were excluded. Finally, 13 articles were used in qualitative synthesis, and reported quantifiable outcomes.

### 3.2. Studies’ Characteristics

All of the selected 13 articles were published in English, and no paper was published before 2004. Three papers were based in Asia (one in Japan and two in South Korea) [22,23,24], and one paper was based in Brazil [25]. All of the other nine papers were conducted in Western countries (one in each of England, France, Australia, Finland, Republic of Ireland, Canada, and USA, and two in Germany). The age group of the included intervention studies was between 0 and 18 years old. With the exception of water fluoridation in three papers [22,26,27], the duration of the other studies varied from 19 months to 6 years. The follow-up time of four articles was 19 months to 2 years [16,17,19,25], and the other six articles had a longer duration time, between 4 and 6 years. Three of the studies had a larger sample size that included more than 3000 participants [15,24,28], whereas the other nine studies had more than 400 and less than 2000 participants. Although one study did not report the number of participants, 47 prefectures were included as the prefecture-level of that study [23].

The included studies used different socioeconomic schema to define socioeconomic position (SEP). One study used blue and white collar as different socioeconomic groups [18], and another study classified one-parent and two-parent family as indicator of SEP [16]. Nine studies used income/education/occupation/employment to define different socioeconomic position, and two studies defined deprived or non-deprived by using area or postcode [20,28]. Dental caries was used as an outcome in this systematic review. Only one paper assessed decayed, extracted, and filled teeth or surfaces (deft/s) for detecting decayed teeth [15], and two studies did not report which criteria were used in assessing dental caries [16,24]. The other studies analyzed decayed, missing, and filled teeth or surfaces in primary (dmft/dmfs) or permanent dentition (DMFT/DMFS). The included studies used different criteria in measuring the same outcome. Five studies assessed dental caries based on World Health Organization (WHO) criteria [19,22,24,26,27], three studies used International Caries Detection and Assessment System (ICDAS) criteria in assessing decayed teeth [17,20,25], and one article used the British Association for the Study of Community Dentistry (BASCD) to assess dental caries [28].

The included studies reported the use of different interventions for the prevention of caries. Four papers used fluoride toothpaste, fluoride mouth-rinse, or dental sealants as interventions [17,23,24,28]. Oral health education or oral health promotion strategies were explored as preventive strategies in five papers [5,18,19,20,25]. Another four papers discussed the effectiveness of water fluoridation [15,22,26,27]. The effect of different interventions on inequality in reducing dental caries is shown in Table 1.

#### 3.2.1. Water fluoridation

Among the four papers that assessed the impact of water fluoridation, one study did not have a control group, but compared inequalities before and after cessation of water fluoridation program, and showed that area-based deprivation inequalities increased after cessation of the program [15]. The other three papers used intervention assessment for the whole sample that included two different preventive groups [22,26,27]. Two studies compared a community water fluoridation (CWF) group and non-CWF group [22,26]. In another study, two communities with different oral health prevention strategies were included [27]. Water fluoridation showed a positive impact on reducing inequality in dental caries in these four studies [15,22,26,27].

#### 3.2.2. Topical fluorides

Fluoride products were used as intervention in four studies. One study assessed the impact of fluoride gel, fluoride toothpaste, and fissure sealant [17], and one study conducted dental sealants as intervention [24]. Another study conducted a school-based fluoride mouth-rinse program in the whole population [23]. These three studies showed a positive preventive effect on dental caries, which reduced the socioeconomic difference in caries distribution. The fourth study found that fluoride toothpastes did not reduce deprivation-related inequality in oral health [28].

#### 3.2.3. Oral health promotion programs

Five studies conducted oral health education to teachers or parents. These studies provided oral health promotion material and oral health knowledge or telephone counseling for parents or teachers as intervention [5,18,19,20,25]. One study conducted both motivational interview as intervention and conventional education as control group [25]. The results of the included studies were inconclusive. The oral health promotion program was effective in reducing the prevalence of dental caries and improving oral health in higher socioeconomic status, thus increasing inequality in dental caries in two studies [18,19]. Two studies showed that the intervention reduced the prevalence of dental caries in lower SEP groups, decreasing inequalities in dental caries [16,25]. Another study concluded that the oral health promotion program has no effect on inequality in dental caries [20].

### 3.3. Quality of Included Studies

Figure 2 shows the risk of bias for the outcomes within each study. All the studies were defined as low risk of bias in incomplete outcome data, selective reporting and other bias, and unclear risk of bias in allocation concealment. Low risk of bias was identical in random sequence generation among eight studies, and high risk of bias existed in one paper [18]. One study presented low risk of bias in blinding of participants and outcome assessment [25]. Four studies had low risk of bias and one study was at high risk of bias in blinding of personnel [18]. The rest of the studies did not have enough information to assess the risk of bias for these attributes.

## 4. Discussion

In this systematic review, two independent investigators searched the literature pertaining to inequality in dental caries, and 13 intervention studies in different Asian and Western countries were identified. There were variations in the quality of included papers, particularly in terms of intervention groups, classification of socioeconomic position, blindness, criteria of caries diagnosis, and outcomes. Five studies regarded oral health promotion as intervention, four papers assessed fluoride products, and another four studies assessed water fluoridation. Variations in the length of follow-up and sample size were also common limitations. The findings of this systematic review indicated that interventions targeting the whole population are more likely to reduce socioeconomic inequalities in dental caries among children.

Water fluoridation and the use of fluoride containing products result in caries reduction [29]. Dental caries is significantly greater among ethnic minorities, people living in rural areas, and socially disadvantaged children [30,31]. However, studies conducted in lower socioeconomic areas have shown that fluoride use was effective in preventing caries development, as it reduced the incidence of caries [32,33,34]. Children in a higher socioeconomic position might already have good oral health behaviors, such as better utilization of dental services, higher frequency of toothbrushing, and fluoride toothpaste, while those in a lower socioeconomic position are less likely to practice better oral health behaviors [28]. Hence, the provision of fluoride is more likely to be useful among those in a lower socioeconomic position.

Five included studies that targeted the whole population showed consistent findings, which showed that the utilization of fluorides is useful in reducing inequality in dental caries [15,22,23,26,27]. Two studies for a target population had a positive effect on reducing inequality in caries only between intensive intervention and no intervention groups [17,24]. However, Ellwood’s study indicated that providing a high concentration of free fluoride toothpaste did not reduce deprivation-related inequality [28]; this could be attributed to the pattern of use of toothpaste among different socioeconomic groups, regardless of the availability of the toothpaste. Whole population interventions, mostly using community water fluoridation and fluoride mouth rinse, showed a greater and consistent impact on reducing inequalities compared with target population interventions. It is worth noting that some have argued the whole population interventions might increase inequalities, as indicated by a media campaign for the prevention of smoking [35]. However, according to Rose’s theory [36], such whole population interventions are superficial as they just encourage people to change their behaviors without acknowledging their underlying circumstances [37]. On the other hand, more radical whole population interventions such as banning smoking or water fluoridation are more likely to reduce inequalities [37].

To further emphasize the impact of population-based intervention/policies, a study modeling the impact of sugar taxation depicted that such policy is anticipated to reduce inequalities in dental caries [38]. Although intake of sugars is the main determinant of the disease, no intervention study used sugar restriction as a way to reduce dental caries.

It is worth noting that other intervention studies conducted in deprived populations were effective in reducing caries, however, the reduction of inequalities in dental caries was not assessed in these studies [39,40,41].

While oral health education programs are usually evaluated using different parameters such as acquisition of knowledge or change in oral hygiene [9], the current review only focused on change in inequalities in dental caries and overlooked other parameters of these programs as they were deemed irrelevant to the objective of the review. In fact, some have argued that oral health education programs are more effective among children in a higher socioeconomic position, and less effective among those in a lower SEP [11,42], and hence increase inequalities. Two included studies demonstrated this observation, where inequalities in dental caries were increased [18,19]. This could be attributed to the unequal allocation of oral health resources among different socioeconomic groups. The cost of oral health products is a barrier for lower SEP groups. Moreover, the attitudes and standard of oral health education among parents and teachers with a higher SEP are more likely to lead them to better understand and implement innovations relevant to oral health promotion among children, which results in better effectiveness of the preventive program in higher socioeconomic groups [19].

However, some studies contradicted this view. One included study found that the oral health promotion program resulted in a greater reduction in the frequency of dental caries among children with a lower SEP [16]. However, the benefits of the oral health promotion program among the lower SEP group could be attributed to the already high caries lesion among them at baseline, with the risk of caries being four times greater than those with high SEP. In other words, there was greater reduction in caries among low SEP groups, but inequality persisted. In Silva’s study, motivational interview showed a greater impact on reducing caries among the lower income group compared with conventional oral health education [25]. This could be explained by the nature of motivational interviews with a face-to-face conversation that focuses on participants’ motivation, their reality, and the barriers that prevent them from adopting healthy behavior. On the other hand, the comparison was with conventional health education, which is known to increase inequality.

Intervention studies related to child oral health have aimed to reduce childhood caries by encouraging children to establish and maintain effective oral health routines [9]. Providing early preventive programs to children in low-income families is critical to achieve and maintain good oral health throughout life [43]. Importantly, interventions that do not exert an additional financial burden or ignore the priorities of those with adverse socioeconomic circumstances are more likely to succeed in reducing inequalities.

This systematic review of the literature has a few limitations. Firstly, although grey literature was considered, only published articles were included in this review, which could contribute to publication bias. There is also publication bias in publishing positive results, which could have impacted the results of this review. Secondly, the overall quality of included papers was considered moderate, given that some of the included studies did not confirm blinding of participants or providers, or allocation concealment. However, this is inevitable in whole population interventions. Finally, meta-analysis could not be performed owing to the high heterogeneity of the included papers.

Oral health promotion programs, particularly those that implement fluoride intervention, are undoubtedly effective in preventing dental caries among children, but they would not necessarily reduce inequality. Indeed, some oral health promotion programs, particularly oral health education and behavior change, might increase caries inequalities [9,10,18,19]. The limited number of studies that demonstrated a reduction in socioeconomic inequalities in dental caries identified in this review highlights two points. First, population-based interventions that do not require prioritizing options for setting resources or daily activities by those subjected to adverse socioeconomic circumstances are more likely to impact inequalities. However, some of these whole population interventions are often hindered by political wills and the priority for setting resources at the governments level. Second, these oral health promotion interventions should be accompanied by other fiscal, societal, and environmental policies that aim at reducing socioeconomic inequalities.

## 5. Conclusions

This systematic review included published intervention studies aiming at reducing inequality in dental caries among children. The overall findings of the included intervention studies imply that whole population intervention is more consistent in reducing inequalities. The included studies varied in measurements of interventions, sample size, age groups, and follow-up time. Larger randomized controlled trials aiming at reducing socioeconomic inequalities in oral health are needed to support or refute the relationship.

## Figures and Tables

**Figure 1 ijerph-18-01300-f001:**
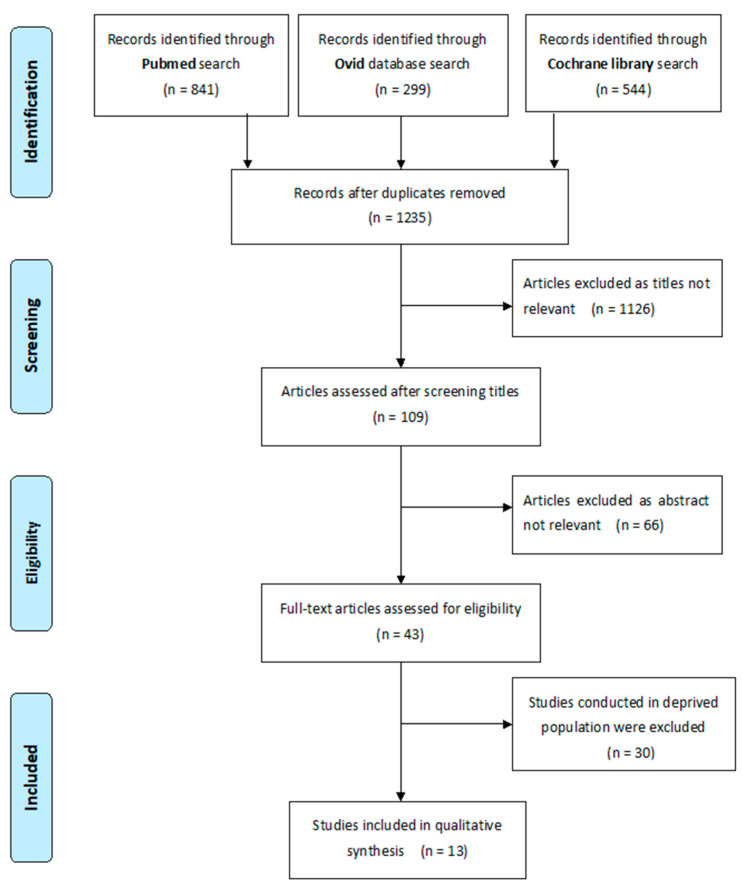
Flowchart of the selection of studies for the systematic review.

**Figure 2 ijerph-18-01300-f002:**
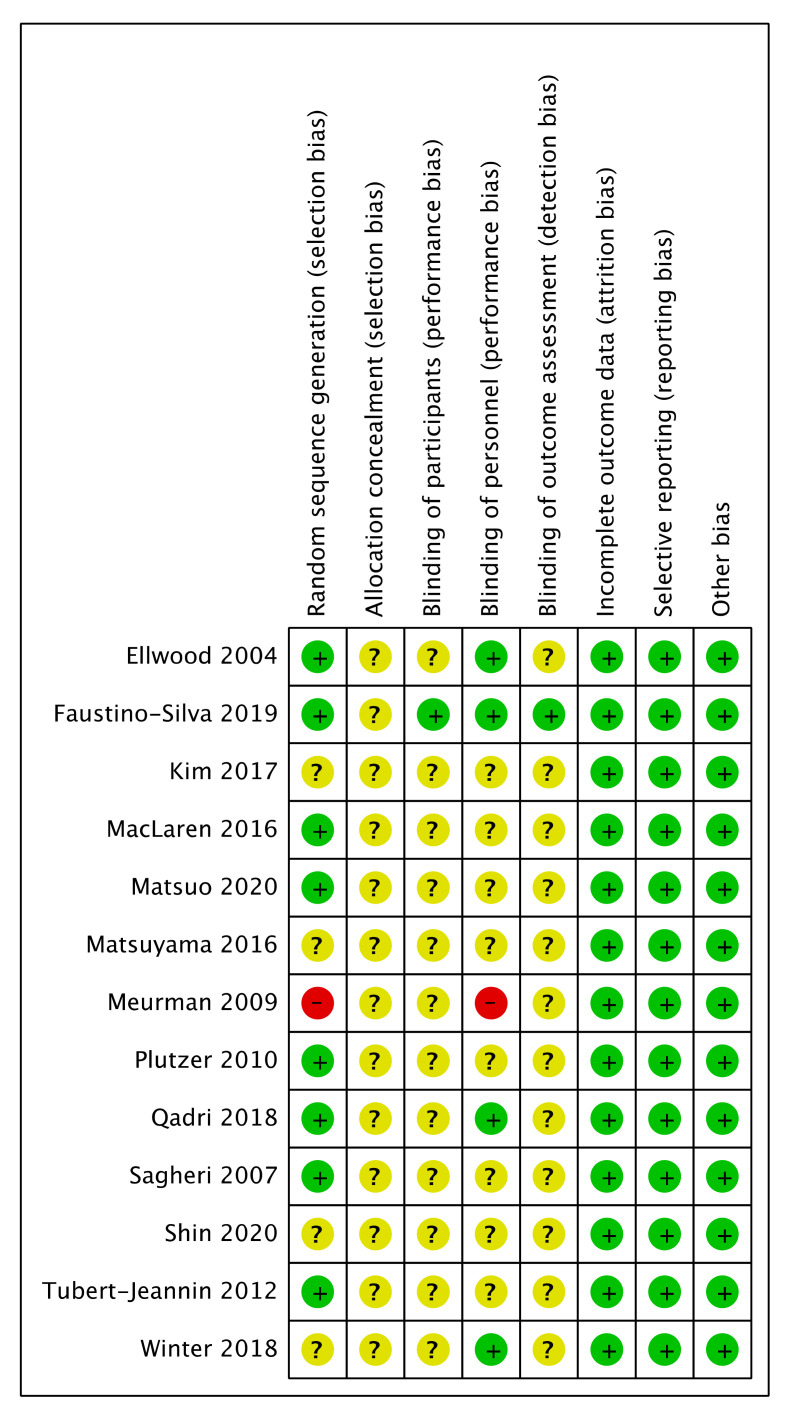
Risk of bias assessment.

**Table 1 ijerph-18-01300-t001:** The effect of different interventions on inequality in dental caries.

Interventions	Author/Year	Intervention	Result
Oral health promotion programs	Meurman (2009) [18]	Oral health education	Increased inequality
	Qadri (2018) [19]	Oral health education	Increased inequality
	Plutzer (2010) [16]	Oral health promotion	Decreased inequality
	Faustino-Silva (2019) [25]	Motivated interview (intervention group)/conventional education (comparison group)	Decreased inequality
	Tubert-Jeannin (2012) [20]	Oral health education	No effect on inequality
Topical fluoridates	Matsuyama (2016) [23]	Fluoride mouth-rinse program/fluoride toothpaste	Decreased inequality
	Winter (2018) [17]	Fluoride gel/fluoride toothpaste/fissure sealant	Decreased inequality
	Shin (2020) [24]	Dental sealants	Decreased inequality
	Ellwood (2004) [28]	Fluoride toothpaste	No effect on inequality
Water fluoridation	Sagheri (2007) [27]	Water fluoridation	Decreased inequality
	McLaren (2016) [15]	Water fluoridation	Decreased inequality
	Kim (2017) [22]	Water fluoridation	Decreased inequality
	Matsuo (2020) [26]	Water fluoridation	Decreased inequality

Increased inequality: intervention increased inequalities in dental caries; decreased inequality: intervention reduced inequalities in dental caries; no effect on inequality: no effect on inequality in dental caries.

## Data Availability

The data presented in this study are available on request from the corresponding author.

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
