# Peer review of "Systematic Review of Intervention Studies Aiming at Reducing Inequality in Dental Caries among Children"

_ijerph, 2021, doi:10.3390/ijerph18031300_

Round 1

Reviewer 1 Report

I read with great interest the manuscript entitled: "Systematic review of intervention studies aiming at reducing inequality in dental caries among children," and the topic is very interesting. The study is appropriate and may be suitable for being published in IJERPH.

This systematic review assessed the evidence on interventional programs reducing inequality in dental caries in the young population. The manuscript is well written, and it is clearly exposed. Nevertheless, some very slight suggestions should be made:

KEYWORDS:

  • Should appear in alphabetical order.
  • For keywords, where possible, please use Medical Subject Headings terms (MeSH Terms).

REFERENCES:

  • Please check the format in All references. It is wrong. Follow the instructions from IJERPH: Author 1, A.B.; Author 2, C.D. Title of the article. Abbreviated Journal Name Year, Volume, page range.

Author Response

We would like to thank the reviewer for your comments. We prepared point by point response.

I read with great interest the manuscript entitled: "Systematic review of intervention studies aiming at reducing inequality in dental caries among children," and the topic is very interesting. The study is appropriate and may be suitable for being published in IJERPH.

This systematic review assessed the evidence on interventional programs reducing inequality in dental caries in the young population. The manuscript is well written, and it is clearly exposed. Nevertheless, some very slight suggestions should be made:

KEYWORDS:

  • Should appear in alphabetical order.

Response to the reviewer: We have changed this. “Child, Dental caries, Inequalities, Socioeconomic Factors, Systematic review”.

  • For keywords, where possible, please use Medical Subject Headings terms (MeSH Terms).

Response to the reviewer: I modified the key words, and used MeSH Terms.

REFERENCES:

  • Please check the format in All references. It is wrong. Follow the instructions from IJERPH: Author 1, A.B.; Author 2, C.D. Title of the article. Abbreviated Journal NameYearVolume, page range.

Response to the reviewer: I updated the references.

Reviewer 2 Report

The novelty of the Systematic review is not highlighted clearly in the introduction.

The introduction part is not very well written. It needs to be revised. Could you please example there are differences between the earlier published paper and the present manuscript?

The conclusions should be implemented

The introduction provides a good, generalized background of the topic that quickly gives the reader an appreciation of the wide range of problems about a dental caries. However, to make the introduction more substantial, the author may wish to provide several references to substantiate the claim made in the first sentence (that is, provide references to other groups who do or have done research in this area). 

For example in the first sentencens you say “ Dental caries is considered one of the most prevalent chronic diseases worldwide “ why don't you write the percentage of the world population who suffer from it?  How is this percentage distributed around the world ?

However, to make the motivation clearer and to differentiate the review some more from other applied papers, the author may wish to provide another sentence giving examples of some of the ideas along with appropriate references. 

I think the rationale for this study needs to be made clearer. In particular, the connection between social inequalities and tooth decay may be clearer. One way to demonstrate this connection would be to cite references (if possible) that show for example how in some highly industrialized countries the use of fluoride can decrease the incidence of caries. 

In the Section Discussion you write: “This systematic review of the literature has few limitations. Firstly, although grey literature was considered, only published articles were included in this review, which could contribute to publication bias. There is also publication bias in publishing positive  results, which could have impacted the results of this review. Secondly, some of the included studies did not confirm blinding of participants or providers, or allocation concealment. Finally, meta-analysis could not be performed due to the high heterogeneity of  the included papers. “ and in Conclusions: “This systematic review included published intervention studies aiming at the reducing inequality in dental caries among children. The overall findings of the included intervention studies imply that whole population intervention is more consistent in reducing  inequalities. The included studies varied in measurements of interventions, sample size, age groups and follow-up time. Larger randomized controlled trials aiming at reducing socioeconomic inequalities in oral health are needed to support or refute the relationship.  

These statements may lead the reader to think "What then is the purpose of this review?" 

The literature cited is relevant to the study, but the authors makes assertions without substantiating them with references. There are also some very old references I suggest updating them to newer ones 

Author Response

We would like to thank the reviewer for your comments. We prepared point by point response.

The novelty of the Systematic review is not highlighted clearly in the introduction.

  1. The introduction part is not very well written. It needs to be revised. Could you please example there are differences between the earlier published paper and the present manuscript? The introduction provides a good, generalized background of the topic that quickly gives the reader an appreciation of the wide range of problems about a dental caries. However, to make the introduction more substantial, the author may wish to provide several references to substantiate the claim made in the first sentence (that is, provide references to other groups who do or have done research in this area). 

Response to the reviewer: Thank you for raising this point. We have now modified the introduction to add more references. (Page2 Line58-61) ‘There are few systematic reviews or reviews that assessed the effectiveness of fluoride vanish or oral health promotion interventions in preventing dental caries (Timms, 2020; Sousa, 2019; Silva, 2016). However, there is no previous systematic review of intervention studies aiming at reducing inequality in caries among children. The current review aims at filling this gap in the literature.’

  1. For example in the first sentences you say “ Dental caries is considered one of the most prevalent chronic diseases worldwide “ why don't you write the percentage of the world population who suffer from it?  How is this percentage distributed around the world?”

Response to the reviewer: We have amended the text and add the following sentence. “The percentage of children among disadvantaged families who suffered from early childhood caries was reported as 70% (Gomez, 2020).” (Page 1 Line37-39)

  1. However, to make the motivation clearer and to differentiate the review some more from other applied papers, the author may wish to provide another sentence giving examples of some of the ideas along with appropriate references. 

Response to the reviewer: Thank you very much for this comment, we have added the following sentence: ‘There are few systematic reviews or reviews that assessed the effectiveness of fluoride vanish or oral health promotion interventions in preventing dental caries (Timms, 2020; Sousa, 2019; Silva, 2016). However, there is no previous systematic review of intervention studies aiming at reducing inequality in caries among children. The current review aims at filling this gap in the literature.’ (Page2 Line58-61)

  1. I think the rationale for this study needs to be made clearer. In particular, the connection between social inequalities and tooth decay may be clearer. One way to demonstrate this connection would be to cite references (if possible) that show for example how in some highly industrialized countries the use of fluoride can decrease the incidence of caries. 

Response to the reviewer: We have added the following sentences: “Dental caries is significantly greater among ethnic minorities, people living in rural areas and socially disadvantaged children (Clark, 2020; Dye, 2012). However, studies conducted in lower socioeconomic areas have shown that fluoride use was effectiveness in preventing caries development as it reduced the incidence of caries (Muller-Bolla, 2013; Slade, 2011; Zaror, 2020).” (Page 8 Line 217-221)

In the discussion, we also added the following sentences: (Page 8 Line 221-225)

“ Children in higher socioeconomic position might already have good oral health behaviours, such as better utilization of dental services, higher frequency of toothbrushing and fluoride toothpaste, while those in lower socioeconomic position are less likely to practice better oral health behaviours. Hence, the provision of fluoride is more likely to be useful among those in the lower socioeconomic position.”

  1. In the Section Discussion you write: “This systematic review of the literature has few limitations. Firstly, although grey literature was considered, only published articles were included in this review, which could contribute to publication bias. There is also publication bias in publishing positive results, which could have impacted the results of this review. Secondly, some of the included studies did not confirm blinding of participants or providers, or allocation concealment. Finally, meta-analysis could not be performed due to the high heterogeneity of the included papers. “ and in Conclusions: “This systematic review included published intervention studies aiming at the reducing inequality in dental caries among children. The overall findings of the included intervention studies imply that whole population intervention is more consistent in reducing inequalities. The included studies varied in measurements of interventions, sample size, age groups and follow-up time. Larger randomized controlled trials aiming at reducing socioeconomic inequalities in oral health are needed to support or refute the relationship.”

These statements may lead the reader to think "What then is the purpose of this review?" The literature cited is relevant to the study, but the authors makes assertions without substantiating them with references.

Response to the reviewer: Thank you for this comment. The purpose of this systematic review is to identify intervention studies aiming at reducing inequality in dental caries among children.

We identified whole population interventions, mostly using community water fluoridation and fluoride mouth rinse, as the interventions with greater and consistent impact on reducing inequalities compared to target population. (Page 8 Line 232-240)

We have added additional references in the introduction and in the discussion (see previous point). We have also added the following sentence in the discussion: “It is worth noting that some have argued the whole population might increase inequalities, as indicated by media campaign for prevention of smoking. However, according to Rose’s theory, such whole population interventions are superficial as they just encourage people to change their behaviours without acknowledging their underlying circumstances. On the other hand, more radical whole population interventions such as banning smoking, or water fluoridation are more likely to reduce inequalities.”

Reviewer 3 Report

I appreciate the opportunity to present the arguments on which I base myself for the acceptance of the reviewed article titled “Systematic review of intervention studies aiming at reducing inequality in dental caries among children” (manuscript ijerph-1065927) to be published in IJERPH. 

The document entitled "Systematic review of intervention studies aiming at reducing inequality in dental caries among children" shows an interesting work related to studies that show intervention programs aimed at reducing inequality in childhood dental caries in economic and social groups of different countries. The results are based on the analysis of a vast bibliography selected considering well-established criteria and yield very conclusive results.

The objective of this work is to present a review of works published up to December 2020 about to intervention programs focused on reducing caries in children up to 18 years of age, and in identifying whether socioeconomic aspects, as well as the intervention of community programs, help to mitigate or not the impact of these programs on the incidence of caries in socially different groups. The relevant aspects of this study focus on water fluoridation, topical fluorides, oral health promotion programs, and they take as reference works from different countries, which makes this research different from that reported in other works. The articles are selected based on the methodology proposed methodology, so they are adequate for the reported parameters. According to the proposed methodology, the choice of articles was carried out based on the objective set and based on well-defined selection criteria. The results are based on 13 articles and presented in a very descriptive way). The results, and the analysis, present the qualitative systematic review (without meta-analysis due to the heterogeneity of the included articles, according to the authors). In my opinion, the conclusions are consistent with the evidence, and arguments presented, and fulfill the objective of the work.  

However, I could suggest the following to the authors 

  1. Abstract. Authors must indicate the number of articles reviewed, the database, and the keywords used. Also how many articles were considered for the study? 
  2. The introduction is very general, raises some aspects that are related to the presence of caries, referring to only a few articles. These aspects were the focus of the research of the present work. It would be desirable that the authors refer to the references review on which the research is based. For example: 

Line 54-60 de la page 2: “While a few studies found intervention programs that reduced inequality in dental caries among children, others have argued that intervention programs increased or had no effect on inequality…….”  

There are no references to support the statement. 

  1. Authors mention that: …….“The quality of included papers was moderate….”…. .Explain in detail that’s mean moderate, and how it is related to the information in figure 2. Give more detail.  

Why do the authors do not search for higher quality articles for their analysis? 

  1. In Table 1, the authors could indicate the reference number corresponding to this according to the list of references
  2. Minor corrections of ingles are required 

Author Response

We would like to thank the reviewer for your comments. We prepared point by point response.

I appreciate the opportunity to present the arguments on which I base myself for the acceptance of the reviewed article titled “Systematic review of intervention studies aiming at reducing inequality in dental caries among children” (manuscript ijerph-1065927) to be published in IJERPH. 

The document entitled "Systematic review of intervention studies aiming at reducing inequality in dental caries among children" shows an interesting work related to studies that show intervention programs aimed at reducing inequality in childhood dental caries in economic and social groups of different countries. The results are based on the analysis of a vast bibliography selected considering well-established criteria and yield very conclusive results.

The objective of this work is to present a review of works published up to December 2020 about to intervention programs focused on reducing caries in children up to 18 years of age, and in identifying whether socioeconomic aspects, as well as the intervention of community programs, help to mitigate or not the impact of these programs on the incidence of caries in socially different groups. The relevant aspects of this study focus on water fluoridation, topical fluorides, oral health promotion programs, and they take as reference works from different countries, which makes this research different from that reported in other works. The articles are selected based on the methodology proposed methodology, so they are adequate for the reported parameters. According to the proposed methodology, the choice of articles was carried out based on the objective set and based on well-defined selection criteria. The results are based on 13 articles and presented in a very descriptive way). The results, and the analysis, present the qualitative systematic review (without meta-analysis due to the heterogeneity of the included articles, according to the authors). In my opinion, the conclusions are consistent with the evidence, and arguments presented, and fulfill the objective of the work.  

However, I could suggest the following to the authors 

  1. Authors must indicate the number of articles reviewed, the database, and the keywords used. Also how many articles were considered for the study? 

Response to the reviewer: Thank you for your comments. The abstract now includes number of articles reviewed, the database, and the keywords used (highlighted).

“Abstract: Background: The objective is to systematically review the evidence on intervention programs aiming at reducing inequality in dental caries among children. Methods: Two independent investigators searched MEDLINE, Cochrane library and Ovid up to December 2020 to identify intervention studies assessing impact on socioeconomic inequalities in dental caries among children. The interventions included any health promotion/preventive intervention aiming at reducing caries among children across different socioeconomic groups. Comparison groups included children with alternative or no intervention. Cochrane criteria was used to assess interventional studies for risk of bias. Results: After removal of duplicate studies, 1235 articles were retained. Out of 43 relevant papers, 13 articles were identified and used in qualitative synthesis, and reporting quantifiable outcomes. The included studies varied in measurements of interventions, sample size, age groups and follow-up time. Five studies assessed oral health promotion or health-education, four assessed topical fluorides and four assessed water fluoridation. Interventions targeting the whole population showed consistent reduction of socioeconomic inequalities in dental caries among children. Conclusion: The quality of included papers was moderate. High heterogeneity did not allow aggregation of the findings. Overall findings suggest that whole population interventions such as water fluoridation are more likely to reduce inequalities in children’s caries than target population and individual interventions.

Keywords: Dental caries; Inequalities; Child; Systematic review”

  1. The introduction is very general, raises some aspects that are related to the presence of caries, referring to only a few articles. These aspects were the focus of the research of the present work. It would be desirable that the authors refer to the references review on which the research is based. For example: 

Line 54-60 de la page 2: “While a few studies found intervention programs that reduced inequality in dental caries among children, others have argued that intervention programs increased or had no effect on inequality….” There are no references to support the statement.  

Response to the reviewer: I’ve added references where required. (Page 2 Line 62-64)

“While a few studies found intervention programs that reduced inequality in dental caries among children (Plutzer, 2010; Silva, 2019; Winter, 2018; MacLaren, 2016), others have argued that intervention programs increased or had no effect on inequality (Meurman, 2009; Qadri, 2018; Jeannin, 2012)”.

References were already in the manuscript.

  1. Authors mention that: …….“The quality of included papers was moderate….”…. .Explain in detail that’s mean moderate, and how it is related to the information in figure 2. Give more detail.  

Response to the reviewer: We would like to thank the reviewer for raising this point. In the results section (Page7 Line 198-205), we stated what different studies reported about risk of bias across different indicators. Given that several studies did not report risk of bias for factors such as blinding, concealment of allocation, etc, the overall quality is considered moderate. We have added an additional sentence in the discussion (Page 9 Line 281-283).

Why do the authors do not search for higher quality articles for their analysis? 

We did conduct comprehensive search of intervention studies, and we reported what we found. Higher quality papers do not exist. Furthermore, in population-based intervention, these type of bias are inevitable.

  1. In Table 1, the authors could indicate the reference number corresponding to this according to the list of references

Response to the reviewer: The list of references was updated.

  1. Minor corrections of ingles are required 

Response to the reviewer: I have edited the paper and updated the errors.

Round 2

Reviewer 2 Report

no comments